# Risk and Protective Environmental Factors Associated with Autism Spectrum Disorder: Evidence-Based Principles and Recommendations

**DOI:** 10.3390/jcm8020217

**Published:** 2019-02-08

**Authors:** Leonardo Emberti Gialloreti, Luigi Mazzone, Arianna Benvenuto, Alessio Fasano, Alicia Garcia Alcon, Aletta Kraneveld, Romina Moavero, Raanan Raz, Maria Pia Riccio, Martina Siracusano, Ditza A. Zachor, Marina Marini, Paolo Curatolo

**Affiliations:** 1Department of Biomedicine and Prevention, Tor Vergata University of Rome, 00133 Rome, Italy; siracusanomartina@hotmail.it; 2Child Neurology and Psychiatry Unit, Systems Medicine Department, Tor Vergata University of Rome, 00133 Rome, Italy; gigimazzone@yahoo.it (L.M.); ariannabenvenuto@yahoo.it (A.B.); rominamoavero@hotmail.com (R.M.); curatolo@uniroma2.it (P.C.); 3Division of Pediatric Gastroenterology and Nutrition, Massachusetts General Hospital for Children, Harvard Medical School, Boston, MA 02114, USA; afasano@mgh.harvard.edu; 4Hospital General Universitario Gregorio Marañón. 28009 Madrid, Spain; alicia.alcon@iisgm.com; 5Division of Pharmacology, Utrecht Institute for Pharmaceutical Sciences, Faculty of Science, Utrecht University, 3584 Utrecht, The Netherlands; a.d.kraneveld@uu.nl; 6Child Neurology Unit, Neuroscience and Neurorehabilitation Department, Bambino Gesù Children’s Hospital, IRCCS, 00165 Rome, Italy; 7Braun School of Public Health and Community Medicine, Hebrew University, Hadassah 99875, Israel; razraanan@gmail.com; 8Child and Adolescent Neuropsychiatry, Federico II University, 80133 Naples, Italy; piariccio@gmail.com; 9Department of Biotechnological and Applied Clinical Sciences, University of L’Aquila, 67100 L’Aquila, Italy; 10The Autism Center/ALUT, Assaf Harofeh Medical Center, Sackler Faculty of Medicine, Tel Aviv University, 69978 Tel Aviv, Israel; dzachor@bezeqint.net; 11DIMES, Bologna University, 40126 Bologna, Italy and IRCCS Fondazione Don Carlo Gnocchi, 20148 Milan, Italy; marina.marini@unibo.it

**Keywords:** Autism Spectrum Disorder, risk factors, protective factors, environment, genetics, medications, toxicants, recommendations

## Abstract

Autism Spectrum Disorder (ASD) is a complex condition with early childhood onset, characterized by a set of common behavioral features. The etiology of ASD is not yet fully understood; however, it reflects the interaction between genetics and environment. While genetics is now a well-established risk factor, several data support a contribution of the environment as well. This paper summarizes the conclusions of a consensus conference focused on the potential pathogenetic role of environmental factors and on their interactions with genetics. Several environmental factors have been discussed in terms of ASD risk, namely advanced parental age, assisted reproductive technologies, nutritional factors, maternal infections and diseases, environmental chemicals and toxicants, and medications, as well as some other conditions. The analysis focused on their specific impact on three biologically relevant time windows for brain development: the periconception, prenatal, and early postnatal periods. Possible protective factors that might prevent or modify an ASD trajectory have been explored as well. Recommendations for clinicians to reduce ASD risk or its severity have been proposed. Developments in molecular biology and big data approaches, which are able to assess a large number of coexisting factors, are offering new opportunities to disentangle the gene–environment interplay that can lead to the development of ASD.

## 1. Introduction

Autism Spectrum Disorder (ASD) is a complex biological condition characterized by a common set of behavioral features with early childhood onset, reflecting the interaction between different genetic and environmental risk factors [1]. 

At present, there is no ultimate treatment for the core features of ASD. Nevertheless, autistic symptoms can be reduced by early behavioral interventions [2,3], and some pharmacological therapies are available for the treatment of psychiatric comorbidities [4].

ASD prevalence seems to be increasing: most recent estimates suggest a prevalence of 1 in 59 among 8-year-old children from the USA (https://www.cdc.gov/ncbddd/autism/data.html) [5]. Another study estimated a 3.5 prevalence increase between 2001 and 2011 in 2- to 17-year-old children [6]. What caused this increased prevalence, beyond a broadening of ASD diagnostic criteria and a better ascertainment of cases, is still unclear. Still, as ASD is the final consequence of cascade events impacting brain development from gestation to early postnatal life [7], it is possible that a true rise is related to these complex events.

While the etiology of ASD is not fully understood, genetics is a well-established risk factor [8]. Twin studies suggested a 76% concordance in monozygotic twins, confirming a strong genetic hereditability for ASD, but also supporting an important contribution of environmental factors [9]. 

Genetic defects in more than 100 genes and loci, and hundreds of copy number variants (CNVs) and single nucleotide (SNVs) polymorphisms (SNPs) have been implicated in about 20% of ASD cases [10,11,12,13]. DNA microarrays enable the discovery of rare and recurrent CNVs as important contributors to ASD and lead to gains in the understanding of autism genetics and to the identification of individuals who might be genetically susceptible to autism. Hotspots of recurrent CNVs, including 16p11.2, 22q11.2, 1q21.1, 7q11.23, and 15q11–q13, have been shown to be strongly associated with ASD [14]. Next-generation sequencing (NGS) methods revolutionized ASD gene discovery and have also substantially contributed to functional genetic data, linking mutations frequently associated with ASD with genes involved in the regulation of brain transcriptional networks during brain development and early synaptogenesis, thus throwing some light on the understanding of the neurobiological consequences of the disruption of these ASD-associated genes [12,15]. Nevertheless, also single-genes syndromes have been associated with ASD, including Fragile-X (FMR1), Tuberous Sclerosis Complex (TSC1-2) and PTEN syndrome [16,17]. 

Nonetheless, the heterogeneous clinical and biological phenotypes observed in ASD strongly suggest that, in genetically susceptible individuals, environmental risk factors also combine or synergize to generate a “threshold point” that might determine a dysfunction. While progress has been made towards gaining an understanding of genetic and epigenetic factors, environmental risk factors are less understood [18]. Actually, recent studies have demonstrated that during critical periods of central nervous system development, early exposure to a variety of environmental factors, ranging from microbes (bacteria and viruses) to medications, from chemicals to physical agents, can affect neurobiological development, including effects relevant to ASD [19,20]. 

In October 2018, international ASD experts convened in Rome to discuss the potential pathogenetic role of environmental factors, as well as their interactions with genetic susceptibility, focusing on three biologically relevant windows for brain development: the periconception, prenatal and early postnatal periods. From the epidemiological point of view, the identification of the exact timing of action of each environmental factor, as well as its consequences in the neurodevelopmental pathways, remains elusive. Nevertheless, it is now possible to establish some differentiations among risk factors that can assist in developing detection and personalized follow-up of populations at higher risk for ASD.

In this paper, we summarize the results of this consensus conference and put forward clinical recommendations for clinicians to reduce ASD risk and/or its severity.

## 2. Conception Period

Advanced parental age: The association between older parental age at conception and neuropsychiatric disorders in offspring is now well documented [21,22]. In the case of ASD, both advanced maternal and paternal age at time of birth (≥35 years) were associated with an increased risk of ASD [23,24,25,26]. Emerging evidence also confirms a combined parental age effect, which is highest when both parents are in the older age range and increases with increasing differences in parental ages [27]. Both human and animal model studies support the hypothesis of an association between elevated rates of de novo mutations in older fathers and increased risk of ASD [28,29]. It has been also suggested that maternal mechanisms mediating the effects of advanced maternal age on ASD risk are associated not only with chromosomal or genetic modifications, but also with a higher prevalence of chronic diseases and a less favorable uterine environment, often resulting in more obstetrical complications, which might eventually lead to an increased risk of adverse birth outcomes [26]. 

Use of hormonal induction and/or assisted reproductive technologies: Assisted Reproductive Technologies (ART) now account for 1–3% of all live births in the Western world (https://www.cdc.gov/reproductivehealth/index.html) [30]. Several procedures that are used in the ART process, such as hormonal stimulation, egg retrieval, in vitro fertilization (IVF), intra-cytoplasmic sperm injection (ICSI), micro-manipulation of gametes and exposure to culture medium, could subject the gametes and early embryos to environmental stress and may be associated with an increased risk of birth defects and low birth weight (LBW) [31]. Children conceived using ART are also at a higher risk for congenital anomalies including a two-fold increase in the central nervous system and epigenetic and imprinted disorders [32,33,34]; there is some evidence that ART might have an impact on imprinting through DNA methylation [35]. Actually, assisted conception and ASD share several risk factors. In both cases, hormonal disturbances, especially in testosterone/androgen regulation, along with high rates of advanced parental ages, preterm deliveries, and LBW, have been reported [6,24,36,37].

Additionally, a recent meta-analysis indicated that the use of ART may be associated with a higher risk of ASD in the offspring [31]. In a previous case-control study conducted on a large Israeli population [38], a higher ART prevalence (IVF and ICSI) (10.7%) even in young mothers (<29 years) was reported among ASD children compared to the overall ART rate. In addition, the study ruled out the hypothesis that ART was associated with unique autism symptomatology (i.e. autism severity and adaptive functioning, a history of developmental regression) that may represent a distinct clinical phenotype in this group. The study results indicated that although assisted conception may be a risk factor for ASD, this group did not appear to represent a separate clinical phenotype within the autism spectrum. These findings suggest that the increased recent prevalence of both ART and ASD might be related.

Environmental chemical and toxicant factors: There is some evidence that exposure to chemical pollutants at critical developmental stages may affect neural and behavioral development. The pathogenetic mechanisms of environmental chemical factors can involve neurotoxicity but can also extend to pathways of immune dysregulation, altered lipid metabolism, and mitochondrial dysfunction. To date, the strongest evidence of association is shown by traffic-related air pollutants and pesticides at different times of exposures [39,40]. 

Maternal nutritional status: Maternal nutritional status and body mass index before pregnancy have been considered as environmental factors that can influence normal brain development through excess or deficit of micronutrients and growth factors, which can affect neurodevelopmental outcomes of offspring [41,42]. In this view, both maternal obesity and underweight have been associated with an increased risk of ASD [42,43]. Maternal obesity results in activation of the maternal immune system and in a chronic inflammation of the uterine environment potentiating abnormal neuronal growth and differentiation in the fetus, with consequent neurodevelopmental impairments in the offspring [44]. At the same level, maternal undernutrition may elicit a physiological stress response leading to neuronal damage through a disproportionate release of proinflammatory factors [45].

A large number of recent studies have suggested association between pre-conception intake of folate and risk of ASD onset in newborns [46]. A significantly higher rate of ASD has been found in children not exposed to folic acid (FA) compared to in children of mothers who took it. Conversely, some apparently conflicting results were reported by other studies that related an increased risk for ASD and neurocognitive impairments in children of mothers who used dietary supplements of synthetic FA [47,48,49]. A possible explanation of these diverging results might be offered by the different compositions between the FA used in supplements (pteroylmonoglutamic) and the one from natural food sources (ormyl-tetrahydropteroylglutamates). High levels of pteroylmonoglutamic acid, which depend on liver-based metabolism, could result in high levels of unmetabolized and non-useful FA in the blood, which can cause changes in brain synaptic transmission and dysregulation of expression of many genes associated with ASD [50,51,52].

Another important micronutrient potentially linked to the neurodevelopmental alterations in ASD is iron. The importance of a correct intake of iron is evident already from the peri-conception period [53]. In the brain, iron contributes to neurotransmitter production, myelination and immune function. In this view, iron deficiency in this period could result not only in impairment in the general development of cognitive, motor and language skills, but also in deficit in social orientation and engagement that could lead to ASD [53]. 

Medications: A growing number of researches highlighted the potential association of prenatal exposure to Selective Serotonin Reuptake Inhibitors (SSRIs) with the onset of ASD, hypothesizing a pathogenetic link between alterations in serotonin pathways and ASD neurobiological abnormalities [54,55,56]; exposure during the preconception period or the first trimester seems to be associated with a higher risk compared to the other two trimesters [57]. Others have found that antidepressants, regardless of their composition, might be associated with increased ASD risk [58]. Thus, some diverging results have been found in relation to both antidepressant types and dosages [58,59]. Furthermore, a Danish longitudinal study, with a follow-up of 5,057,282 person-years, did not detect a significant association between maternal use of SSRIs during pregnancy and ASD in the offspring [60]. Moreover, another large research did not find, after controlling for several confounding factors, a significant association between prenatal exposure to antidepressant medication and ASD [61]. Lastly, a “confounding by indication” cannot be excluded, raising the possibility that it is depression and anxiety that might be risk factors for offspring ASD, rather than antidepressants per se [58]. Nevertheless, even if evidence is still conflicting, the recommendation is to proceed to apply the precautionary principle, balancing the use of antidepressants against the substantial adverse consequences of untreated maternal depression. 

## 3. Prenatal Period

Environmental chemicals and toxicants: In the last few years, epidemiologic investigations indicated that prenatal exposure to chemical and toxic factors such as air pollution, pesticides, materials used in the plastic industry and heavy metals may increase the risk of ASD [39,40,62,63,64]. Possible mechanisms behind the association between these environmental risk factors and ASD are not only their interactions with genetic factors, and/or epigenetic marks leading to a diminished ability to detoxify xenobiotics [65,66] but also their potential role in triggering neuro-inflammation and oxidative stress that lead to neurobiological and neurotransmitter alterations and abnormalities in signaling pathways [63]. 

Air pollution: Air pollution is probably the chemical risk factor with the strongest evidence of association with ASD, especially for exposures in the third trimester [40,67,68]. Multiple variables, such as metrics of exposure, type of pollution, time of exposure, could influence the risk of ASD and its clinical outcome [69,70,71]. It should be noted, however, that despite positive associations that were observed in many countries like the USA, Canada, Taiwan and Israel, European studies did not find any association [71,72,73]. Recently, in a Canadian population-based birth cohort, a significant association between exposure to nitric oxide and ASD was found. No association was found between ASD and particulate matter with a diameter of <2.5 μm or nitrogen dioxide [74]. These contradictory results might be due to the fact that studies of air pollution have been often limited by indirect and cross-sectional methods of exposure measurement, by different metrics of exposure, by different evaluations of outcomes, and by focusing on different pollutants. Notwithstanding the need for further investigation, and even if some unanswered questions remain, prenatal air pollution exposure has emerged as a potentially modifiable risk factor for ASD. 

Pesticides: Exposure to organochlorine pesticides (measured using geographical mapping) increases the risk of offspring ASD [75,76]. Additionally, studies that examined risk of ASD in relation to prenatal levels of poly-chlorinated biphenyls (PCBs) reported a suggestive association with specific PCBs [20,77,78]. Organophosphate exposure during pregnancy increases the risk of autistic symptoms in the offspring, at 2–3 years of age [79]. In particular, residential proximity to organophosphates at some point during pregnancy is associated with a 60% increased risk for offspring to develop ASD [80]. Conversely, a pilot case-control study investigating risk associated with exposure to organic pollutants (including a variety of PCB congeners, DDT - dichloro-diphenyl-trichloroethane, and DDE - Dichloro-2,2-bis(p-chlorophenyl)-ethylene) measured in archived maternal serum and diagnosis of ASD in children did not find significant differences in odds ratios for ASD [77]. Most pesticides of current use are neurotoxic, may target the developing brain [81] and are prone to cause oxidative stress [82]. The widely used pyrethroids have been associated to ASD and neurodevelopmental delay [83]. Nevertheless, for the same reasons described before, when talking about air pollution, an association between pesticide exposure and ASD is not yet confirmed [78]. 

Phthalates: Phthalates are a class of chemicals used as plasticizers, solvents, and lubricants, and as enteric coatings on pharmaceuticals and nutritional supplements. Few studies have addressed the relationship between ASD and prenatal exposure to phthalates (3rd trimester) with contrasting results [78].

Heavy metal exposure: Little evidence for an association between hair metal concentration of mercury, copper, cadmium, selenium, chromium and autistic symptoms has emerged until now [84]. Moreover, as most of these studies only measured biomarkers and do not ascertain actual exposure sources, temporality of association is unknown. Some studies examined exposure in relation to maternal dental amalgam fillings and maternal or child consumption of seafood with inconsistent findings [19]. A meta-analysis found not only consistent evidence for lack of association between childhood thimerosal exposure and ASD, but also an increased risk of ASD following a higher level of inorganic mercury exposure [85]. 

Medications: The association between ASD and prenatal exposure to drugs is increasingly investigated; a specific area of interest was the study of antiepileptic and antidepressant agents [54]. 

Among antiepileptic drugs (AED), valproate showed the strongest association with neurodevelopmental outcome, in terms of cognitive disabilities, developmental delay, and ASD [86]. It is therefore contraindicated as a first-line antiepileptic or mood stabilizer in pregnant women or in those who plan pregnancy. Moreover, other AED, as oxcarbazepine and lamotrigine (alone or combined with valproate), have been found to be associated with the onset of ASD in the offspring [87]. Findings across several meta-analyses examining the association between antidepressant exposure during pregnancy and ASD are reasonably consistent showing an increased risk [88]. Additionally, maternal psychiatric disorders could play a critical role in the development of ASD; thus, these have been considered also as a potential confounding or addictive risk factor for exposure to antidepressants alone [88,89,90]. 

Some studies suggested also a possible link between prenatal or early-life antibiotic use and ASD [91], but too limited information is currently available to draw conclusions. Recently, however, it has been demonstrated that low-dose antibiotic exposure in late pregnancy and early postnatal life in mice induces impaired social behaviors and aggression in mice associated with changes in the intestinal microbiome [92]. On the other hand, supplementation with the probiotic Lactobacillus Rhamnosus JB-1 might prevent the early-life antibiotic-induced aberrant behaviors. Taken together, these results merit further research on the potential role of early-life antibiotic exposure in the development of ASD.

Substance abuse: A large number of studies examined prenatal exposure to substance abuse as heavy tobacco smoke, alcohol, or cocaine and ASD. Association between high amounts of alcohol consumption in pregnancy and ASD in offspring (especially those with Fetal Alcohol Syndrome) is documented [93,94,95]. On the other hand, association between moderate alcohol intake in pregnancy and ASD is unlikely [94]. 

An association between smoking during pregnancy and risk of childhood autism has been suggested [96], but in this case, results are conflicting, with two meta-analyses in a total of 15 studies reporting no association with overlapping odds ratios [97,98]. Therefore, at present, insufficient data have been found to support an association. 

Nutritional factors: Epidemiological studies and data obtained in humans have provided evidence that mother’s diet during pregnancy plays a critical role in the development of the neural circuitry that regulates behavior, thus determining persistent behavioral effects in the offspring [48]. Generally, it is known that some elements of maternal diet during pregnancy, such as FA, vitamin D, iron and fatty acids, are associated with higher or lower incidence of ASD or autistic traits in the offspring [99]. Specifically, low concentrations of vitamin D and FA are associated with an increased risk of ASD diagnosis, in particular if these deficiencies are present in the mid-gestational period [100,101]. In addition, a maternal diet with high levels of methanol and aspartame during gestation could be linked to an increased risk of ASD [102]. 

A poor omega-3 intake during gestation and maternal high-fat diet during pregnancy has been associated with the risk of ASD and other neurodevelopmental disorders [19,103]. In fact, high-fat consumption during pregnancy is strongly associated with activation of several of the same inflammatory cytokines (e.g., interleukins IL-4, and IL-5) that are elevated during gestation in mothers of children with ASD. Furthermore, high-fat diet consumption in pregnant women is associated with modifications of the neural pathways involved in behavioral regulation, specifically the serotoninergic system. The suppression of serotoninergic synthesis in the brain may underlie the risk of developing later behavioral disorders, as long as the offspring is exposed to maternal high-calorie diet during pregnancy. 

Prenatal infections and maternal immune activation: Current data suggest that at least for a subset of women, exposure to infections during pregnancy might increase ASD risk or other disorders of the central nervous system (CNS) in the offspring. Activation of the maternal immune response can confer a risk for the onset of psychiatric disorders. In particular, exposure to prenatal infections, such as flu, rubella, measles, herpes simplex virus, and bacterial infections, may increase the risk for the offspring of developing bipolar disorder and schizophrenia [104]. More recently, some population-based cohort studies described a potential link between autism risk and maternal infection or inflammation during pregnancy, depending on the time of gestational exposure, the type of infective agent, and the intensity of the maternal immune response; specifically, viral infections seem to be associated to ASD risk in the first trimester, bacterial infections in the second trimester, influenza and febrile episodes during the whole pregnancy but especially in the third trimester [105,106]. Fewer studies have examined the potential impact on ASD risk of fever as such, rather than in connection with infection broadly [106]. A retrospective case-control study based on maternal self-report showed an association between fever during pregnancy and increased ASD risk [105]; it showed also that this risk was attenuated only in mothers who took anti-pyretic medications to control their fever, but not in those mothers who did not [105]. A prospective study in Norway also found an increased risk for ASD after prenatal fever exposure, as well as evidence of a dose–response relationship, with risks rising parallel to multiple episodes of maternal fever [107]. 

A prevailing concept is that maternal immune activation (MIA) may alter the expression of inflammatory molecules in the developing fetus and that maternal-fetal immune dysregulation may disrupt brain development and neural connectivity, which in turn may have long-term effects on the offspring’s mental functions [108]. Among the studies supporting a link between maternal infection and increased risk of ASD, there are several ones carried out with the quantification of cytokine, chemokines and of other inflammatory mediators measured in the maternal serum and amniotic fluid [109]. These studies, however, have generated conflicting results [56,105]. Recently, increased levels of maternal cytokines and chemokines during gestation have been associated with subsequent ASD with intellectual disability [110].

Maternal immune systems can be involved in increasing ASD risk, even independently from prenatal infections. In particular, maternal autoantibodies might recognize proteins in the developing fetal brain [111]; these autoantibodies can be detected in ~20% of mothers of children at risk for developing autism versus 1% of mothers of typically developing children, and defined an additional sub-phenotype of ASD [112,113]. 

Individual maternal factors and diseases: Gestational diabetes has been considered a risk factor because it negatively affects fetal growth and it increases the rate of pregnancy complications [114,115,116]. Moreover, it impacts long-term fine and gross motor development and leads to learning difficulties and attention-deficit hyperactivity disorder [117]. These adverse effects of maternal diabetes on brain development may arise from the increased fetal oxidative stress, as well as from epigenetic changes in the expression of several genes [114,115,118]. However, the increased risk for ASD linked to gestational diabetes may be related to pregnancy complications rather than to complications secondary to hyperglycemia. Whether control of diabetes reduces ASD risk is still unknown [114,115].

Additionally, maternal melatonin levels have been investigated as potential culprits in the ASD pathogenesis [119]. Melatonin is a crucial hormone for neurodevelopment and protects from oxidative stress and neurotoxicant agents. Melatonin deficiency is frequently detected in ASD children already in a very early period of life, and thus the potential implications of low maternal melatonin levels have been considered as a factor that might increase the susceptibility to autism [120].

## 4. Perinatal/Early Postnatal Period

Current research seems to suggest that obstetric risk factors occur more often in ASD children compared to neurotypical controls, even though these results have been challenged by other authors [121]. In this view, the higher prevalence of obstetric negative events in ASD could be explained, not only by the maternal genetic/epigenetic mechanisms mentioned above, but also by hormonal factors altering the in utero environment, leading to a fertility decline and increasing pregnancy and obstetric complications, which lead to emergencies, such as caesarean sections (CSs) or preterm births [114].

Several studies have examined the possible relationship between CS and/or induced labor and ASD, with conflicting findings [122,123]. One of the pathogenetic hypotheses is the possible effect of oxytocin (OT) variations during CS in the etiology of ASD. Epigenetic dysregulations of the oxytocinergic system could play a role in the behavioral dysfunctions of ASD. Perinatal alterations of OT can also have life-long lasting effects on the development of social behaviors [124]. Within the perinatal period, various processes, like planned caesarean section, labor induced by synthetic OT or interrupted with oxytocinergic antagonists, can also alter the OT balance in the newborns, even though the implications and medium/long-term effects of these manipulations are still largely unknown [123,125].

Other studied perinatal factors include gestational age of <36 weeks, spontaneous, induced, or no labor, breech presentation, as well as preeclampsia and fetal distress [24,26,126]. In preterm births, chorioamnionitis, acute intrapartum haemorrhage, and LBW have been associated with higher risks of abnormal results during early autistic screenings [127]. According to a study, parity of ≥4 might be a protective factor that decreases ASD risk [126]. 

Microbiome: Scientific evidence is beginning to accumulate suggesting that, within ASD populations, the gut microbiome shows a different composition compared to typically developing individuals (e.g., higher representation of *Clostridia*, *Bacteroidetes*, *Desulfovibrio*, and *Sutterella* spp), which might be responsible for frequent gastrointestinal disorders experienced by patients with ASD. Recent evidence in ASD subjects suggests that microbiota transplantation could represent a promising approach to improving gastrointestinal and ASD symptoms [128]; data, corroborated also by ASD animal models, showed the potential beneficial effects of probiotics treatment and fecal microbiota transplantation [129]. However, further research is necessary in order to evaluate the effective long-term improvements on the ASD clinical phenotype. Another way to target the microbiome is dietary intervention with prebiotics, including fibers, such as galacto-oligosaccharides (GOS) that induces the growth and activity of beneficial bacteria [130]. Recently, it has been suggested that the combination of exclusion diets and GOS supplementation might result in significant improvements in anti-social behavior in ASD [131].

While the fetal environment was initially thought to be entirely sterile, recent evidence suggests that some bacteria are present in the amniotic fluid and placenta. One implication of these new discoveries is that the microbial composition of the developing offspring may be sensitive to environmental changes even during prenatal stages of life [132]. Animal models suggest that maternal gut bacteria can promote neurodevelopmental abnormalities in offspring, possibly mediated by T-helper-17 cells with subsequent immune system activation [133].

Table 1 provides a detailed summary of the main hypothesized environmental risk factors implicated in ASD in relation to their most important period of exposure.

## 5. Gene–Environment Interactions and Epigenetics

The pathogenetic role of environmental risk factors in ASD etiology must not be considered as a separate element but rather like a complex network of factors that can epigenetically affect genetic components. Furthermore, recently, there has been an emphasis in shifting from associative observations to more mechanistic studies to establish cause–effect relationship linking environmental factors to ASD pathogenesis [26].

For example, an association between ozone exposure and ASD risk has been demonstrated only among individuals who have a high CNV burden [61]. Moreover, the impact of MIA on the onset and severity of ASD seems to be significantly influenced by genetic susceptibility [137,138]. Some environmental factors such as certain toxins and vitamin D deficiency increase the risk of gene mutation that, in turn, can lead to an increased risk of ASD [139]. Likewise, as an association between maternal adiposity and variations in newborn blood DNA methylation has been confirmed, it could lead to the modified expression of several important genes (such as *apolipoprotein D*) that are critical to neurodevelopment in utero [135].

To explain the effects of gene–environment relations, many researches pointed their interest in examining potential involvement of epigenetics in ASD etiology [140,141]. Epigenetic mechanisms are biochemical modifications of DNA that affect gene expression without changing the DNA sequence; these are influenced by exposure to environmental factors [20]. Epigenetic programming is dynamic and responsive to different environmental exposures during development and includes several interrelated processes, including chromatin remodeling, histone modifications, DNA methylation, and expression of microRNAs (miRNAs). Epigenetic mechanisms play a critical role in normal brain development, thus drawing a bridge between genetic predisposition and environmental factors. They affect brain functions throughout the whole life, both at the individual and the transgenerational levels [140]. Several studies examined also the putative effects of stressful experiences in utero, such as prenatal infections, in epigenetic processes. In particular, animal models of MIA revealed that prenatally infected offspring exhibited significant differences in the expression of miRNA, altered histone modification, and changes in DNA methylation [132].

## 6. Protective Factors

As already outlined, our knowledge about possible genetic and environmental risk factors for ASD is improving. However, the individual developmental trajectories and outcomes are not just the result of the influence of risk factors, as the interaction between risk and protective factors has to be considered as well. In the past, protective factors were thought to be just those characteristics inherent to the individual, such as a high intellectual quotient or better social skills. Now, there is an increasing understanding that there is the need to look also at possible pre- and postnatal environmental factors. However, despite the growing interest in the identification of environmental risk factors and their potential prevention in order to decrease ASD risk, to date, little is known about protective factors for ASD. Nevertheless, in the last few years, increasing efforts have been made to try to identify factors that may improve long-term outcomes [142]. 

Some elements of the mother’s diet might play a protective role by countering some core autistic symptoms. The main elements of the maternal diet that seem to play a protective role against ASD are fatty acids, vitamin D (vit. D), and iron [99].

A mean daily FA intake of ≥600 μg in the periconception period and/or during the first month of pregnancy, but only in cases of significant mother’s fatty acids deficiency, is associated with a 40% decrease of ASD risk [100,143]. The association between fatty acids and reduced ASD risk is strongest for mothers and children with MTHFR 677 C > T (cytosine > thymine) variant genotypes, which leads to less efficient folate metabolism [143].

According to some studies, vitamin D supplements during pregnancy could reduce the risk of developing ASD in the offspring [144].

Higher iron intake through the end of pregnancy and particularly during breast feeding was associated with reduced ASD risk compared to lower intakes [53]. 

In addition, some meta-analyses provide evidence that breastfeeding (exclusively or accompanied by additional supplements) may protect against ASD [145]. Breastfeeding may reflect the protective effect of breast milk [145,146,147]; for example, breast milk contains bifidobacteria, lysozyme, lipoxins, glutathione, and anti-inflammatory cytokines. Literature suggests that, relative to controls, children with ASD have lower levels or bifidobacteria and lysozyme in the digestive tract and increased levels of inflammatory cytokines in plasma. Therefore, a number of possible components of breast milk could plausibly be connected to a decreased ASD risk.

Additionally, the interest in polyunsaturated fatty acids (PUFA) in maternal diet is increasing, as lipid composition (lipidomics) seems crucial in psychiatric disorders [148]. The increase in PUFA, especially omega-3 fatty acids, in a prenatal maternal diet was associated with a decreased ASD risk [19].

Another factor that could play a role as a protective agent is melatonin. Melatonin synthesis is frequently impaired in patients with ASD and in their mothers. Therefore, consumption of this hormone during pregnancy could act as a neuroprotective factor, decreasing the risk of neurodevelopmental disorders, including ASD [120].

Table 2 summarizes the main known protective factors for ASD, in relation to the suggested period of exposure. These are likely protective factors that might prevent and/or modify a poor outcome in a more positive one, thus enhancing the potential for a healthier life. For all these reasons, focus on reinforcing protective factors should be increasingly considered a key element in the preventive method as well as in the clinical approach to ASD.

## 7. Early Intervention Strategies

The preconception and prenatal periods are probably the stages, in which the risk and protective factors play their major role. However, the well-known postnatal plasticity of the brain suggests that, during the first month and/or year of life, there are additional prospects to mitigate the impact of ASD on the quality of life of the affected person. In fact, developmental trajectories in children with ASD are complex and highly variable, so one of the major challenges is identifying potential protective factors and developing effective treatments. Single-gene syndromes with a high prevalence of neurodevelopmental disorders such as TSC or PTEN syndrome provide a unique opportunity to investigate risks and protective factors. Actually, the development of the phenotype does not stop when the diagnosis is made; the several risk and protective factors persist in acting together across the whole life, but particularly during the earliest stages. For this reason, early detection and subsequent early intervention strategies might positively modify the evolving developmental trajectories. 

Actually, beyond some general environmental aspects, which have been shown to be protective, such as an inclusive educational environment or a positive parenting, in the last few years, a number of peripheral markers have been identified in children affected by idiopathic autism, including altered redox balance and mitochondrial dysfunction [150], decreased DHA and cholesterol with impaired Na^+^/K^+^-ATPase activity in erythrocyte membranes [151,152], upregulation of inflammatory cytokines and dysfunctional microbiota [153,154], and a characteristic metabolomic signature [136,155]. These altered parameters not only may provide tools for early diagnosis, but should be regarded as hubs of a network of inter-related dysfunctions, which are the basis for the manifestation of autistic clinical symptoms and its more common comorbidities. Providing remediation for such peripheral dysfunctions, specifically in a very early period of life, may make a significant difference in lessening the severity of symptoms, thus improving the quality of life. Ideally, one should monitor in high-risk infants and in newly diagnosed toddlers and children, a number of biological parameters, including nutritional deficiencies, oxidative stress, mitochondrial dysfunction, inflammation markers, intestinal leakage markers, and lipid composition of membranes, and then provide appropriate nutraceutical supplementations. Following a few months of treatment, the clinical outcome and the biological parameters should be assessed, in order to evaluate the efficacy of treatment. 

## 8. Clinical Recommendations

Table 3 summarizes the main clinical recommendations for each period of vulnerability. However, as extensively discussed in the previous chapters, the impact in terms of ASD risk of several of these factors is only partially clarified; thus, these clinical recommendations should be acknowledged considering possible pitfalls related to still existing contradictory data interpretations. For this reason, these recommendations should be pondered in the perspective of the precautionary principle. Some general recommendations are, of course, related to antenatal care for each pregnancy, such as no smoking, no alcohol consumption, and prevention of folate acid and vitamin D deficiencies. Nevertheless, pediatricians should pay attention also to environmental conditions, which are recognized as risk factors for ASD. These include, but are not limited to, advanced parental age, a sibling with ASD, prematurity, a history of ART, maternal diabetes, and maternal obesity, as well as the use of antiepileptic or antidepressant drugs. In all these cases, close monitoring aiming to minimize the effects of risk factors and to maximize the impact of protective factors is warranted. 

Of course, there are situations, like car exhaust-related or other sorts of air pollution, in which the reduction of the impact of a risk factor is not prerogative of single individual’s choice; rather it is matter of governments’ policies. Nevertheless, some simple strategies, namely ventilation improvements and air cleansers, can be of assistance in reducing at least indoor pollution.

Anyway, it should also be mentioned that several of these factors have been associated with other neurodevelopmental or psychiatric disorders as well, for example with Attention Deficit Hyperactivity Disorder (ADHD), conduct problems, or lower behavioral scores [166]. As a matter of fact, it is now recognized that co-occurrence of neurodevelopmental disorders is more often the rule than the exception. In this perspective, one should ask not just if an environmental factor increases the risk for ASD, but also if it might impact individually just one of the different features of ASD, like social-communication or repetitive behavior.

In short, the synergic effects of genetic, epigenetic and environmental factors can lead to a higher susceptibility during the whole pregnancy, especially in a subset of mothers at high risk of having a child with ASD. Although it is not possible, at present, to change the pathogenetic effects of the majority of these factors, some modifiable environmental agents could be modulated in order to restrain the severity of the disorder and, potentially, to prevent its onset. 

Figure 1 summarizes the impact of selected environmental factors on ASD risk.

## 9. Future Perspectives

Future studies will need to take into account the complexity and heterogeneity of ASD, aiming at detecting the interactions of the diverse and multiple risks and protective factors associated with ASD. Thus, we should move from the research of a single risk factor to models which take into account the dynamic relationship between genetics and environment. Large long-term genetically informed prospective studies, including multi-generational ones, are needed to take into account new genetic/epigenetic evidence, as well as data arising from cellular, computational, or animal models. Large and heterogeneous sample sizes are needed to be able to identify timing of exposure in relation to critical developmental periods in which the risk and protective factors are acting. Furthermore, studies on environment toxics have been until now largely limited by a lack of reliable exposure measurements. In addition, this aspect would benefit from prospective designs.

In conclusion, recent developments in molecular biology and big data approaches, which are able to assess a large number of coexisting factors, are offering new opportunities to disentangle the gene–environment interplay that can lead to the development of ASD.

## Figures and Tables

**Figure 1 jcm-08-00217-f001:**
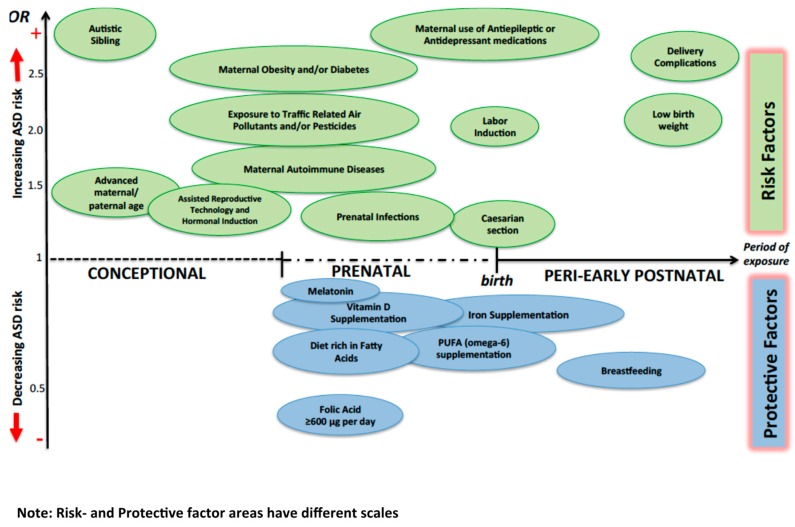
Possible impact of different environmental factors on ASD risk. The Odds Ratio (OR) for protective factors other than breastfeeding usually refers to conditions where the mother presents a lack of the specific factor. OR of “Autistic sibling” is around 7, so it is presented out of scale.

**Table 1 jcm-08-00217-t001:** Possible environmental risk factors for Autism Spectrum Disorder (ASD).

Risk Factor	Hypothesized Period of Action	Selected Studies
Advanced parental age	Conception	Durkin et al., 2008 [23]; Ben Itzchak et al., 2011 [24]; Geier et al., 2016 [25]; Sandin et al., 2016 [27]; Modabbernia et al., 2017 [26]
Use of hormonal induction; Assisted Reproductive Technologies (ART)	Conception	Auyeung et al., 2009 [36]; Zachor & Ben Itzchak., 2011 [38]; Liu et al., 2017 [31]
Environmental chemicals and toxicants: air pollution pesticides phthalates	Conception, prenatal	Volk et al., 2011 [69]; Becerra et al., 2013 [70]; Rossignol et al., 2014 [40]; Weisskopf et al., 2015 [67]; Gong et al., 2017 [71]; Raz et al., 2018 [73]; Eskenazi et al., 2007 [79]; Cheslack-Postava, 2013 [77]; Shin et al., 2018 [134]; Braun et al., 2014 [78]
Nutritional factors: maternal obesity or undernutrition folates vitamin D deficiency iron deficiency	Conception; prenatal; early postnatal	Georgieff et al., 2007 [45]; Krakowiak et al., 2012 [41]; Getz et al., 2016 [42]; Andersen et al., 2018 [43]; Schmidt et al., 2011, 2012 and 2017 [46,135,136]; Vinkhuyzen et al., 2017 [101]; Schmidt et al., 2014 [53]
Medications: valproate other AEDs SSRIs antibiotics antibiotic	Prenatal	Roullet, et al., 2013 [86]; Veroniki et al., 2017 [87]; Mezzacappa et al., 2017 [57]; Atladottir, 2012 [91]
Infections; Fever; Maternal Immune Activation (MIA)	Prenatal	Zerbo et al., 2013 [105]; Jiang et al., 2016 [137]; Brucato et al., 2017 [106]; Zerbo et al., 2013 [105]; Parker-Athill et al., 2010 [109]; Jones et al., 2017 [110]
Maternal individual factors and diseases: gestational diabetes; maternal melatonin levels; depression (?)	Prenatal	Gardener at al., 2009 [114]; Lyall et al., 2012 [115]; Jin et al., 2018 [120]
Delivery method	Perinatal	Dodds et al., 2011 [122]; Emberti Gialloreti et al., 2014 [123]
Fetal distress	Perinatal	Modabbernia et al., 2017 [26]; Wang et al., 2017 [126]

**Table 2 jcm-08-00217-t002:** Proposed protective factors for ASD.

Nutritional Protective Factors	Period of Exposure	Study
Folic acid of ≥600 μg Folic acid + MTHFR 677 C > T variant genotype	Prenatal Prenatal	Schmidt et al., 2012 [143]; Schmidt et al., 2011 [149]; Suren et al., 2013 [100]
Fatty acid PUFA	Prenatal	Lyall et al., 2013 [19]; Morgese et al., 2016 [148]
Vitamin D	Prenatal	Stubbs et al., 2016 [144]
Iron Iron + Breastfeeding	Prenatal; postnatal	Schmidt et al., 2014 [53]
Melatonin	Prenatal	Jin et al., 2018 [120]
Breast feeding	Postnatal	Bar et al., 2016 [146]; Boucher et al., 2017 [147]; Tseng et al., 2017 [145]

**Table 3 jcm-08-00217-t003:** Clinical Recommendations for the periconception, prenatal and early postnatal periods.

Clinical Recommendations	Minimizing Risk Factors	Maximizing Protective Factors	References
Periconception Period	Encouraging women weight loss in case of obesity and strict glycaemia control in case of diabetes; close monitoring and/or treatment of preconception maternal diseases and/or conditions (psychiatric conditions, vitamin D or folic acid deficiencies); close follow-up of children born after ART use using frequent developmental surveillance after birth	Monitor diet of women; encourage assumption of daily folic acid and vitamin D intake from natural sources before pregnancy; have reasonable exposure to sunlight.	Peretti et al., 2017 [99]; Schmidt et al., 2012 [143]; Andersen et al., 2018 [43]; Oberlander et al., 2017 [90]; Zachor & Ben Itzchak, 2011 [38]
Prenatal Period	Close monitoring and symptomatic treatment even for mothers with minor infections or inflammatory episodes; prevention of infections during pregnancy with vaccination programs; surveillance of mothers who are using long-term medications. Mothers who had already autistic children and/or with de novo or inherited ASD-associated CNVs are more susceptible to environmental insults in the subsequent pregnancy; therefore, a strict surveillance and treatment of infections or inflammatory episodes during whole pregnancy is highly recommended.	Recommend daily folic acid intake of ≥600 μg during the first month of pregnancy; recommend a constant intake of vitamin D and iron	Babenko et al., 2015 [156]; Schmidt et al., 2014 [53]; Mezzacappa et al., 2017 [57]; Veroniki et al., 2017 [87]
Perinatal/Early Postnatal Period	Close monitoring not only of premature newborns, but also of those with minor perinatal complications; defined medical and neuropsychological follow-up of preterm children; ASD screening in all preterm infants, as recommended by AAP, using instruments such as M-CHAT In case of syndromic ASD: early and frequent neurodevelopment assessment to promptly identify early signs suggestive of ASD (i.e. deficits in social communication behaviors in TSC, low adaptive behaviors in social area in FXS, lack of language development in Angelman syndrome, and difficulties in joint attention in preterm infants) In case of high risk for epilepsy, EEG monitoring and immediate treatment of seizures (to minimize the impact on long-term outcome) In all high-risk infants, (genetic syndromes, preterm birth, and familial history) parental education to warrant early referral and parent-mediated intervention	Whenever possible, encourage breastfeeding; monitor diet of infants and toddlers; early targeted behavioral interventions to potentiate cognitive abilities, which can act as protective factors reducing the severity of ASD symptoms	Curatolo et al., 2018 [157]; Peralta-Carcelen et al., 2018 [158]; McDonald et al., 2017 [159]; Tseng et al., 2017 [145]; Peretti et al., 2017 [99]; Zwaigenbaum et al., 2015 [160]; Bonati et al., 2007 [161]; McCary et al., [162]; Jones et al., 2017 [163]
	As early as possible in high-risk infants and in newly diagnosed toddlers/children	Following the evaluation of biological parameters, provide appropriate nutraceutical supplementations	Li et al., 2017 [164]; Adams et al., 2018 [165]

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
