# Peer review of "Risk and Protective Environmental Factors Associated with Autism Spectrum Disorder: Evidence-Based Principles and Recommendations"

_jcm, 2019, doi:10.3390/jcm8020217_

Round 1
Reviewer 1 Report
This is a very nicely organized comprehensive review on the preconception, prenatal, perinatal and early postnatal factors that may increase the risk of ASD in human. while the literature is described briefly in a well written style, there is too little emphasize on the negative studies that reject possible associations. Examples for such an approach are the description of the possible association of antidepressant treatment and ASD. The feeling after reading the description is for a positive association while many meta analyses and large population studies (See for example studies by Heli Malm from Finland) failed to demonstrate such association. I would like to see a more balanced description of the data and a through discussion of the possible pitfalls of the data interpretation. Otherwise the paper is well written, the tables are appropriate and the section on protective factors is innovative.
Author Response
Thank you very much for the suggestions. We added, in several parts of the manuscript, a more in-depth description of the different inconsistencies among studies evaluating the role of prenatal exposure to various agents, particularly antidepressants, air pollution, and pesticides.
Reviewer 2 Report
"Risk and protective environmental factors associated with autism spectrum disorder: Evidence-based principles and recommendations" is a meeting report on environmental factors that contribute to ASD. The report is extremely well written, comprehensive, and provides a remarkable synthesis of the current literature. I have only one concern: the lack of consideration of the contributions of maternal antibodies to fetal brain proteins. This work is mostly done by the Judy Van de Water lab at UC-Davis, and has proven to be a substantial contributor to ASD risk. Examples of recent relevant papers, by PubMed ID: 29934547, 27534269, and 27312731. Besides the maternal antibody contribution, the authors provide a comprehensive and fair evaluation of the current literature. The tables and figure are extremely helpful tools for summarizing the literature.
Author Response
Thank you very much for this very important comment. The reviewer is right, we missed to discuss this essential point. We have therefore added in the “maternal immune activation” section, page 6. Thank you again.